# ‘Who Else If Not We’. Medical Students’ Perception and Experiences with Volunteering during the COVID-19 Crisis in Poznan, Poland

**DOI:** 10.3390/ijerph19042314

**Published:** 2022-02-17

**Authors:** Jan Domaradzki

**Affiliations:** Department of Social Sciences and Humanities, Poznan University of Medical Sciences, Rokietnicka 7 St., 60-806 Poznań, Poland; jandomar@ump.edu.pl; Tel.: +48-61-845-27-70 or +48-69-532-46-30; Fax: +48-61-845-27-70

**Keywords:** COVID-19 pandemic, experiences, future healthcare professionals, students volunteering, Poland

## Abstract

Introduction: The first wave of the COVID-19 pandemic resulted in the closure of all Polish medical universities. Simultaneously, due to staff shortages and the Polish health-care system being seriously challenged, many students were eager to contribute to the fight against the outbreak. This study explores medial student volunteers’ (MSV) perspectives and their lived experiences during the COVID-19 pandemic in Poland. Material and Methods: Semi-structured interviews were conducted with twenty-one students. Results: A total of seven major themes emerged from the interviews: 1. students’ reactions to the COVID-19 pandemic, 2. students’ experiences of the outbreak, 3. motivations for volunteering, 4. students’ perceptions of the COVID-19 volunteering, 5. organization of students’ volunteering, 6. benefits and costs of volunteering during COVID-19, and 7. social perception of MSVs. Although students volunteering was an example of civic responsibility and created new learning opportunities, many students felt unprepared for the pandemic, lacked social skills and access to psychological support, and were the target of stigmatization and discrimination. Discussion: Because during the first wave of the COVID-19 pandemic medical universities were closed and classes were held online, students’ volunteering became an important part of service learning and created an opportunity for education. Consequently, while it benefited students, patients and the healthcare system, students’ involvement reinforced such important values of medical ethos as: altruism, public service, and (professional) solidarity. However, some systemic approach should be undertaken that would prepare students better for future crises.

## 1. Introduction

The emergence of the severe acute respiratory syndrome coronavirus 2 (SARS-CoV-2) in December 2019 resulted in the closure of many segments of the national economy, a limitation on travel, the closing of public places, and the implementation of social distancing. However, it was the closure of hospitals and medical universities that seemed to disrupt society the most. Consequently, health-care systems in many countries, i.e., Italy, Spain, or Israel were seriously overburdened by the coronavirus disease 2019 (COVID-19), especially due to staff and personal protective equipment (PPE) shortages [1,2,3,4,5]. Simultaneously, although some countries limited teaching or cancelled medical students’ involvement in medical practices entirely and restricted their contact with patients, others have integrated their education into COVID-19 response systems and have considered enrolling medical students into the workforce, hoping it would prevent dire shortages of healthcare workers. For example, the United States, the United Kingdom, Italy, and China integrated final year students into their healthcare systems or graduated them early so that they could serve as frontline clinicians in the COVID-19 hospitals [6,7,8,9]. In Denmark, medical universities kept their students in clinical placements and initiated fast-track courses in ventilation therapy and nursing assistance [10,11]. In addition, in Germany [12], Poland [13,14,15,16], Vietnam [17], and Indonesia [18], non-final year students were encouraged to volunteer in healthcare facilities.

However, there was uncertainty about the involvement of clinical and preclinical medical students during the pandemic, and their role has been discussed more than ever before [8,19,20]. Some experts argued that because students are not essential healthcare workers, they should focus on studying medicine [21]. It was also suggested that, especially during the first phase of the outbreak, students’ involvement was too risky. Some stressed that students may function as additional vectors for disease transmission and pose a risk for patients and healthcare practitioners alike. Others claimed that medical students would use a great deal of the available PPE, of which there were serious shortages. Finally, it was argued that students were not prepared both in terms of their skills and coping with the possible moral injury caused by the outbreak, and that their involvement would place an extra burden on teaching physicians [7,21,22,23,24,25,26,27].

On the other hand, many insisted on letting students into the hospitals and argued that because healthcare staff are a finite resource and may become depleted as a result of illness, students’ commitment may play a significant role in supporting healthcare workers [9,23,25,28], especially because medical students have supported the medics during previous epidemics, including the Spanish flu outbreak in 1918 in Pennsylvania, or the Polio epidemic in 1952 in Copenhagen [1,3,11]. Some stressed that as future healthcare professionals, students have responsibilities towards patients and should be allowed to fulfil their professional duties [6]. Yet others emphasized that students could assist with non-COVID-19 in-patient care, including respiratory therapy, or outpatient care, assisting the community through providing childcare, grocery collections, or PPE collection drives [6,10,13,14,20,23,29,30]. Finally, it was argued that in contrast to reactivated retired healthcare professionals, students are generally young, healthy, and are a low-risk group [19]. Simultaneously, most authors stressed that medical students’ engagement must be voluntary [31,32].

Similar discussions occurred in Poland, where the first case of the COVID-19 infection was confirmed on 4 March 2020, the first case of the disease in Poznan was confirmed five days later, and the first COVID-19 victim in the country died on 12 March. Consequently, starting from 10 March, various lockdown-type control measures were imposed by the Polish government, including the closure of all universities, which was announced on 11 March. However, following a call from the Ministry of Health, Poznan University of Medical Sciences (PUMS), in collaboration with the university’s student organizations, started inviting its students to volunteer. The reason for this was that although even before the pandemic Poland suffered from healthcare workforce shortages—in 2017, there were 2.4 doctors and 5.1. nurses per 1000 population (OECD mean 3.5 and 8.8, respectively) [33]—COVID-19 has deepened the healthcare staffing crisis and the country’s healthcare system reached its limits. Indeed, from the beginning of the pandemic, 152,436 healthcare professionals in Poland were infected (81,844 nurses, 32,872 physicians, 13,410 physiotherapists, 8416 midwives, 4616 pharmacists, 4116 paramedics, 3986 dentists, 3146 laboratory diagnosticians, and 30 feldshers) and 582 have died (251 physicians, 201 nurses, 61 dentists, 24 midwives, 22 pharmacists, 7 paramedics, 7 physiotherapists, 5 laboratory diagnosticians, and 4 feldshers) [34]. Moreover, as of 29 January 2022, in Poznan and the Wielkopolska region alone, the total number of infections reached 442,172 (12, 66% of population of the region) and 9610 people have died (9, 19% of all infections and 8, 93% of deaths in the country) [35]. Consequently, from the very beginning, the outbreak of the coronavirus disease presented itself not only as a public health emergency but also as a challenge to the health system.

Thus, on 13 March 2020 the rector of PUMS and the governor of the Wielkopolska province signed an agreement which made it possible for volunteers to work in various health care facilities, including hospitals, emergency units, hospital pharmacies, sanitary-epidemiological stations, and the university’s diagnostic laboratory. As a result of this, more than seven hundred students from the faculties of Medicine, Pharmacy, and Health Sciences volunteered during the first wave of pandemic alone.

However, the problem was that in contrast to many other jurisdictions in the Polish law, student volunteering is not regarded as a mandatory form of experiential learning, such as internships. Consequently, although, medical students engage in various forms of volunteering, medical universities in Poland do not regard student volunteering as a part of university curricula, nor it is treated as a way of supporting student learning process [36].

Thus, this study explores medical student volunteers’ (MSV) perspectives and their lived experiences of volunteering during the COVID-19 pandemic. Although knowing the perceptions of frontline healthcare workers is important, understanding medical students’ experiences with the pandemic is also of crucial importance, as it may help us to guide future practice and prepare them better for the next health crisis. In particular, many students felt unprepared for dealing with the pandemic, and for others, volunteering during the COVID-19 was a source of serious burden.

## 2. Material and Methods

### 2.1. Study Design

Although earlier studies focussed on medical students’ attitudes, motivations, and willingness to volunteer during the COVID-19 outbreak, or the role of volunteering in the education of future healthcare professionals and formation of their professional identity, there is still a great shortage of research on the students’ lived experiences during the pandemic. Consequently, this study identifies the facilitators, barriers, and factors affecting medical students’ satisfaction with volunteering during the coronavirus outbreak. Moreover, while most previous studies were designed as quantitative research, this study used qualitative methods [37]. Thus, such in-depth study may help us to better understand the students’ motivations, opinions, and lived experiences. This, in turn, may help us to provide them with the right foundation to learn and thrive, and ensure that medical students will be better prepared when faced with another pandemic.

Thus, semi-structured interviews with students who volunteered during the COVID-19 pandemic in Poznan, Poland, were performed. Because the study focussed on the students’ lived experiences, the meanings they gave to those experiences and the choices they made based on those meanings, an interpretative phenomenological approach was used for this study [38]. The questionnaire was developed according to the Qualitative Pretest Interview (QPI) approach [39]. Thus, the initial list of the interview questions was developed after a thorough analysis of academic literature on the medial students’ volunteering during the COVID-19 pandemic. The structure of the questionnaire was constructed in consideration of the study aim and focussed on medical students’ lived experiences of volunteering during the outbreak. Thus, the interview questionnaire, which consists of 7 categories of questions, was designed to find out what meanings students gave to their experiences as volunteers and how theses meanings influenced their decisions and choices: what were students’ reactions after hearing about the COVID-19 pandemic and the governmental restrictions, why they decided to engage in volunteering and what were the motivations behind their decision, what tasks they performed during their voluntary service, what were their experiences with volunteering during the pandemic, how they rated the organization of students’ volunteering, and what reactions they faced during their voluntary service. Thus, the questionnaire consisted of 32 questions, which facilitated the identification of specific issues related to medical students’ volunteering during the current healthcare crisis (Appendix A).

Before conducting the formal phase of qualitative research, a series of three pretest interviews were conducted to assess the instrumentation rigor and to formulate measures to address any limitations or threats to bias and management procedures. While it helped to reformulate four questions, it also enabled me to identify various obstacles and increase the methodological and social reliability of the questionnaire, which are central considerations to any qualitative research. The final version of the questionnaire was evaluated by three external reviewers: two medical students and one sociologist, and received approval from the University Student Council Board (USCB). Additionally, ethics approval and research governance approval were obtained from the PUMS Bioethics Committee (KB—831/20).

### 2.2. Participant Recruitment and Data Collection

Invitation to participate in the study was posted on an online platform. Students were included if they were directly involved in voluntary service during the pandemic and were eager to participate in the study. The recruitment process was continued until thematic saturation was achieved [40]. All in all, twenty six students responded and agreed to an interview. However, three resigned due to the lack of time and two quit volunteering for personal reasons.

Students were interviewed between January and March 2021. Due to safety reasons, nineteen interviews were conducted as telephone conversations, lasting between 40 and 75 min, and two were performed as written interviews. All interviews were digitally audio-taped and transcribed verbatim. For analytical purposes, any type of emotions, intonations, silences, or emphases were also transcribed.

### 2.3. Data Analysis

The data were analysed thematically, guided by Colaizzi’s approach [41]. First, all transcripts were read multiple times and categorised in a process of familiarisation. Then key words, significant statements, or phrases describing the students’ experiences were sought. The initial results were noted on a separate sheet and assigned preliminary codes. The codes and relevant text excerpts were then consolidated in meaning statements, which were grouped into thematic clusters that were integrated into major themes.

## 3. Results

### 3.1. Participants

A total of 21 student volunteers were interviewed: sixteen females and five males (Table 1). They studied at the following faculties: medicine (*n* = 8), nursing (*n* = 6), rescue medicine (*n* = 2), public health (*n* = 2), pharmacy (*n* = 1), and medical analytics (*n* = 1). Additionally, one PhD student was interviewed. The mean time of voluntary experience during the pandemic was 3.4 months (range: 1.5–12 months). There were ten participants who were already well established students (third, fourth, or fifth year students), while six were first year students. Among the tasks performed during their voluntary service were: helping in local hospitals, emergency units, sanitary-epidemiological stations, or the university’s diagnostic laboratory, where they performed triage, cared for patients, conducted epidemiological interviews over the phone, took the medical history from those infected, helped in administrative work, gave telephone advice in call centres, translated English texts about COVID-19, or helped in the distribution of PPE. A total of sixteen respondents had earlier experiences with various forms of volunteering.

### 3.2. Findings

Overall, seven key patterns emerged during the interviews: 1. medical students’ reactions to the COVID-19 pandemic, 2. medical students’ experiences of the outbreak, 3. motivations for volunteering, 4. medical students’ perceptions of the COVID-19 volunteering, 5. organization of medical students’ volunteering, 6. benefits and costs of volunteering during COVID-19, and 7. the social perception of MSVs (Figure 1).

#### 3.2.1. Theme 1: Medical Students’ Reactions to the COVID-19 Pandemic

While describing their first reactions after hearing about the COVID-19 outbreak and restrictions imposed by the government, students evoked a wide range of emotions. However, the most common reactions were disbelief and fear. Some MSVs described their feelings of uncertainty and anxiety and recalled the apocalyptic visions of previous plagues and omnipresent death fuelled by the media.


*At the beginning it was fear. I thought it would be like a plague and that people would be dying on the streets, especially that the media were constantly showing images from Italy with the soldiers on the streets. There was also rumours that they would close the entire city, so I thought I would be cut off from the world. It was fear and insecurity.*

*(MSV3)*



*There was a lot of adrenaline. I had this feeling that a war was coming.*

*(MSV10)*


Other respondents recalled their mistrust in the government’s assurance that the situation in the country was under control.


*When they [the government—JD] told us that it was going to last for two weeks only, I already knew that it would be much longer, as no epidemic lasts for two weeks.*

*(MSV2)*



*While I was thinking that in relation to the number of infections restrictions were too harsh, I also knew that it wouldn’t be as promising as the minister Szumowski [the Minster of Health—JD] was saying.*

*(MSV8)*


Some students admitted being sceptical about the pandemic, as at the beginning they did not believe it was a real threat and were thinking about it in terms of fear mongering. Simultaneously, they recalled how their perception of the risk has changed over time.


*At the beginning I didn’t believe it was a real pandemic (…) Later when the number grew and the virus appeared in Italy and Poland, I was mad that so many people didn’t follow the restrictions.*

*(MSV15)*



*[i]n October I was very sceptical about it. I thought it would be like in the case of Ebola: they were scaring us that it would also come to Europe but nothing like that happened. When it came to Poland (…) I was scared that Poland would turn into a second Italy.*

*(MSV17)*


Students described their feelings of insecurity, fear, frustration, and anger. However, while some MSVs worried over the negative impact the pandemic might have on society and the healthcare system, most respondents were confused because of not knowing when the crisis would end. They were also frustrated with the restrictions imposed on their personal freedom and were afraid about their future. In particular, they were concerned over the social distancing requirements, which enforced remote education and disrupted their daily activities.


*I was mad and furious. My anxiety over the future mixed with anger that my plans couldn’t be realized. I was sad and mad that my study trip to Summer School abroad was cancelled.*

*(MSV6)*



*I was very upset that all the classes were suspended. I thought it was unnecessary and exaggerated. I was mad that they took away my time from clinical classes.*

*(MSV19)*



*Respondents were also afraid over the health of their loved ones, including parents and grandparents.*



*I was afraid over the health of my parents and grandparents.*

*(MSV12)*



*I was worried that my grandparents may be infected.*

*(MSV20)*


#### 3.2.2. Theme 2: Medical Students’ Experiences of the Outbreak

As the pandemic progressed, all these negative feelings intensified and MSVs expressed concerns over the negative impact of the outbreak on the national economy, healthcare system, and the health of their families. Interestingly, most were not so preoccupied with contracting the virus as they were with the possibility of being asymptomatic carriers and transmitting the virus to others. This was especially the case for those MSVs who were still living with high-risk individuals, such as senior family members with co-morbidities.


*When the pandemic gained momentum I was scared over the health of my loved ones, my family.*

*(MSV11)*



*I was more afraid of being a carrier than of being infected. Because of my age and good physical shape I didn’t feel endangered. I was afraid that I might infect others.*

*(MSV20)*


During the first wave of the pandemic, most MSVs were mainly focused on personal issues and worried about their education and professional futures.


*After two or three months I started wondering how it would affect my education. I worried that we didn’t have access to clinical classes and patients, and that online lectures and seminars were not enough, as we were unable to talk to the patients and examine them.*

*(MSV1)*



*I was afraid that I wouldn’t be able to finish my studies. As we have lost almost a year and a half now, I was wondering how we could catch up and how we would handle ourselves in the future job. I was afraid that we would have to learn everything in practice, at work. It was hard.*

*(MSV15)*


Some MSVs remained stoical and argued that although the situation was difficult, they did not find it particularly disturbing. Moreover, some perceived the pandemic as a “learning opportunity”.


*I didn’t experience any special worries. (…) I thought it was an opportunity for me, a chance to gain extra knowledge.*

*(MSV5)*



*To be honest, I wasn’t scared, nor had any negative feeling about it because I knew that I wasn’t in the group of increased risk. So I tried not to panic. It didn’t affect me.*

*(MSV19)*


However, there were respondents who reported that due to the pandemic and lockdown inconveniences, they experienced serious psychological distress, an increased level of anxiety, and depressive emotions.


*I was scared over my future and health, how I would handle the isolation. Such negative feelings affected me a lot. My relationship broke up, and weeks of isolation resulted in panic attacks.*

*(MSV6)*



*It was social isolation, a prohibition to leave home and an inability to realize my passions that were the hardest experience; and fear over my loved ones. These were the most dominant feelings.*

*(MSV1)*


#### 3.2.3. Theme 3: Motivations for Volunteering

After hearing the news about the COVID-19 pandemic, all MSVs described their need to act and engage in the fight against the outbreak. Although students declared a variety of motives for volunteering, for the majority the prime motivator was the belief that studying at medical university was a unique vocation and that as future healthcare professionals it was their duty to engage and help whatever the risks. Thus, some recalled the feeling of the ‘sublimity of the situation’ and a sense of a ‘mission’ they had, which influenced their desire to be a part of ‘something ground-breaking’.


*I had a feeling of duty. Although I don’t study medicine or pharmacy, I thought that as a public health student I could help. I was thinking ‘Who else if not we’, at the medical university.*

*(MSV4)*



*It may sound naïve, but it was a kind of imperative: you must give something from yourself, especially as you have finished medical university. I wanted to do something, and not be the observer and commentator. I had the sense of vocation.*

*(MSV7)*



*Because I have chosen my studies to help people and I often wondered what would I do if a war started or something I thought that maybe this was the right time to step up. It was a sense of a mission, the feeling of solidarity with the medics and the desire to help the sick.*

*(MSV17)*


Students who had previous experience with various forms of volunteering described their decision as ‘natural’ and declared that as full-fledged volunteers they were driven by the ideal of doing good and helping others, and wanted to give something back to the community.


*For me it was a natural decision, to continue my medical volunteering I have started a year before the pandemic.*

*(MSV5)*



*It wasn’t a difficult decision. I have been engaged in voluntary service from early high school. In high-school I was a member of the Red Cross. I’ve always liked such activity.*

*(MSV6)*


On the other hand, some MSVs were more orientated toward personal goals such as having a personal feeling of satisfaction from helping others, the possibility of passing a summer internship, or simply believing that volunteering was better than sitting at home and studying or being bored. Some also suggested that volunteering helped them to cope with their psychological stress and emphasized its therapeutic dimension.


*Because I come from a small town, I was tired of sitting at home doing nothing apart from walking around or jogging. I wanted to help and engage in something.*

*(MSV15)*



*It had a therapeutic value, because due to the lockdown it was impossible to leave home, meet people or travel. It was difficult; so if I didn’t volunteer it would be an even harder experience, especially that I had moments when I felt helpless and very uncertain.*

*(MSV17)*



*My colleague posted a message that we can pass our summer internships as volunteers. I thought it was a good opportunity to observe how management in a state agency functions. I also wanted to escape a bit from my bad mood.*

*(MSV21)*


Yet others believed that the pandemic was a chance to gain new knowledge and practical skills that could be useful in their future profession.


*I wanted to be closer to what was going on. I wouldn’t gain all that knowledge while sitting at home next to the computer.*

*(MSV4)*



*I thought it was an opportunity: being my age and working during the pandemic. I thought that it may not be the last pandemic and that this experience could benefit me in the future.*

*(MSV5)*


However, most respondents revealed a unique mixture of altruistic and egoistical drives. Thus, even though most were orientated towards altruism and public service, personal enhancement motivations were equally important as they wanted to learn new skills and gain professional experience. However, personal motivations were not the main reason for becoming involved.


*Because I study at this faculty [rescue medicine—JD], I felt, I’ll say it a bit lofty, a moral duty to help. I was also afraid that the healthcare system wouldn’t handle it. I believed that even a freshman like me could help. It was also a kind of adventure. Finally, I was hoping that as our education switched online I could learn something.*

*(MSV8)*



*At the beginning it was the desire to help, but later I was also motivated by the possibility to pass my summer internship.*

*(MSV20)*


Simultaneously, while all students had discussed the decision to volunteer with their parents, families, partners, friends, or teachers, they declared that it was their personal and autonomous choice. However, some suggested that they experienced some forms of external pressure, either from their academic teachers, fellow students, or society.


*There was a kind of social pressure, especially in the media, that we should volunteer.*

*(MSV16)*



*There was a kind of pressure, even from some professors. For example, one professor posted a message on Facebook saying: ‘Either you’d be with us or not’.*

*(MSV17)*


#### 3.2.4. Theme 4: Medical Students’ Perceptions of the COVID-19 Volunteering

Although most MSVs did not have any special expectations related to the voluntary service and often perceived it simply as a ‘job’ or ‘learning opportunity’, some mentioned that before starting it they imagined pessimistic scenarios, which influenced their feelings of security and safety.


*I had this image, like they showed it on the TV, that we would be placed in the military tents at the front of the hospitals, which would serve as a kind of ‘dying rooms’. I was expecting that I might get infected. I knew I was putting myself at risk.*

*(MSV3)*



*I was thinking that (…) we would be sent to covid wards where we would be caring for the sick and monitor their life functions. It made me scared. I thought it might be dangerous.*

*(MSV17)*


Although all students were aware of the risk, it was not the fear of being infected that worried them the most. As already mentioned, most respondents were more afraid of bringing the virus from the hospital to their homes. Consequently, while fearing that they may pose a risk to their loved ones, MSVs felt responsibility for others and avoided contact with their relatives or decided not to go back to their family house. Moreover, some experienced great anxiety for many months, until they were vaccinated.


*I worried that I might infect somebody. I felt a great responsibility that some persons might be hospitalized because of me. Such worries accompanied me during my entire volunteering. Only when we all got vaccinated I felt secure enough I could visit my family at home.*

*(MSV15)*



*Soon after I made the decision I started to worry, and I’ve felt it for months, that I might infect others. My anxiety was intensified by the constant information that young people are mainly carriers.*

*(MSV16)*


Additionally, respondents were afraid that due to their age they lacked the necessary knowledge, skills, and experience, and that they would not be able to perform many tasks.


*Back then I was studying for only six months, so I was worried that my knowledge was vague and that they could assign me things I was not familiar with.*

*(MSV2)*



*I had some worries, especially during my first meeting with the patients. I felt uncertain and tentative as I was not sure whether I was qualified to do all those things. I was anxious that I may not be gentle enough or even harm those people who were already suffering.*

*(MSV14)*


#### 3.2.5. Theme 5: Organization of Medical Students’ Volunteering

MSVs were also asked about the organization of the COVID-19 volunteering. Most respondents argued that the volunteer induction they received was sufficient, as they were trained about all the responsibilities, procedures, and protocols. They also stressed that there was always someone who supported and advised them. Consequently, they felt safe, secure, and prepared for their service.


*We underwent basic training sessions on occupational hygiene, safety procedures and infectious diseases. (…) I was also trained in the patients’ personal data protection. Additionally, there were always persons who were watching and helping us.*

*(MSV2)*



*We underwent sanitary-epidemiological training: how to disinfect our hands and the workplace, wear masks and uniforms. We also knew exactly what to do during our volunteering; and someone was always helping us. I felt very safe and secure.*

*(MSV17)*


However, some respondents complained about not being prepared properly for their functions. While some argued that their training was too short, others claimed that they did not receive training or information whatsoever, and said that students desperately lacked information on their responsibilities. Thus, some argued that they owed their preparation to their previous volunteering or to colleagues who explained everything to them.


*To be honest I felt like being thrown into deep water, without proper preparation. In fact, on the evening before I started someone called me and asked whether I could come the next day. However, I didn’t receive any type of training regarding safety or communication. I simply joined a nurse and was learning from her.*

*(MSV14)*



*We barely had any training and it was my colleagues who showed me everything.*

*(MSV1).*


Although the majority of students declared that their personal safety and protection were taken care of, some complained that at the initial stage of the pandemic they had limited access to PPE. Simultaneously, they were aware that the primary reason for this was the shortages of PPE experienced in most of the countries struggling with COVID-19.


*It was well organized. I felt safe and secure. We were provided with the PPE, although at the beginning there were some problems, as there were only reusable masks. Later they gave us also FFP2s and FFP3s.*

*(MSV5)*



*At the beginning we didn’t have much protective equipment, there were very few masks or disinfectant liquid, but as it [the pandemic—JD] developed we were given the visors and dispensable aprons, and after a month it was much safer.*

*(MSV2)*


Nevertheless, MSVs complained that volunteers’ well-being was not cared for enough as they did not have access to psychological support, especially during the first months when many students experienced stress and emotional burden, caused both by the pandemic and the volunteering itself. Respondents missed institutional support and the possibility of debriefing. Some suggested that it was the volunteers who supported each other.


*Although everyone was very supportive in technical issues, we didn’t have any psychological or emotional support. I missed the possibility of debriefing, and having a chance to share my experiences, anxieties and worries.*

*(MSV6)*



*We didn’t have any type of institutional support, and some persons needed such psychological help, especially those who volunteered at the beginning, as it was a hard, and stressful work. After eight, ten hours on a duty we experienced a psychological and emotional burden.*

*(MSV16)*


Finally, students gave their opinions on the motivational system, which aimed to encourage students’ volunteering during the pandemic. Although most MSVs stressed being motivated by a strong sense of altruism and public service, some admitted that gaining a credit for a compulsory internship was an additional motivator. However, many suggested that it had negatively influenced the quality of volunteers, who were driven more by egoistic drives. Moreover, MSVs complained that as a result, volunteering “lost its spirit”. Additionally, some complained that the knowledge and skills acquired during volunteering were not compatible with those required by the program. There were also students who felt as if they were being used, and stressed that volunteers should receive regular pay.


*I see how it has changed. Now, when students include volunteering as part of their practice their motivations are much different than it used to be when it all began. It has lost its spirit. Back then we were all excited that we were taking part in it, but later many students lacked that commitment. They simply wanted to pass their summer internships. They became more mercenary.*

*(MSV3)*



*Because we do a normal job, I think that students could be offered a regular contract.*

*(MSV5)*


Simultaneously, none of the respondents regretted their decision to join the fight with the pandemic and stressed that the university had made a good decision to call upon students to volunteer. MSVs felt satisfied that during the difficult situation experienced by the Polish healthcare system, they were able to help and relieve the system. Others stressed that it was a kind of a “test” and a chance to prove oneself as future health professionals.


*I think it was a good decision, especially at the beginning when the numbers were skyrocketing. There was a staff shortage and the system needed us.*

*(MSV17)*



*I think it was a good decision, because we are preparing ourselves for a profession which is a kind of sacrifice and it was a chance to prove oneself.*

*(MSV14)*


#### 3.2.6. Theme 6: Benefits and Costs of Volunteering during COVID-19

Even though most MSVs were not involved in any direct patient care activities, all respondents considered their work as useful and believed that it was an important part of their medical education. Moreover, regardless of their age, gender, faculty, initial level of fear, the time of recruitment, or time spent on volunteering, all MSVs expressed a high level of satisfaction from volunteering and stressed the countless benefits that arose from it. Although for some it was a chance to leave home, spend their time in a useful way, make new friends, or pass a summer internship, most respondents emphasized that volunteering made them feel needed and useful, and made them sure that they had chosen the right studies.


*Now I see clearly that these studies have a purpose. Volunteering made me realize how great it is to work for such a great cause. It also helped me to develop self-confidence.*

*(MSV3)*



*The most satisfying was that I had the feeling that I didn’t waste that time. I was useful and I was doing something important. I wasn’t passive.*

*(MSV7)*


Simultaneously, although value-based gratifications were highlighted, some MSVs stressed how volunteering had affected their psychological development and personal growth.


*I could test myself as I didn’t know how would I react in such a critical situation, whether I would go to the front line. Now I know I will, so I have proved this to myself.*

*(MSV10)*



*I felt great satisfaction each time I managed to do something I have learnt during the classes. It was very satisfying when the personnel called for help as they already knew that I could do something and they were treating me as a member of their team.*

*(MSV2)*


Additionally, MSVs emphasized that as medical schools were closed and classes were held online, volunteering created an educational opportunity for learning new things, improving clinical and social skills, and gaining professional experience.


*It gave me a lot of practical knowledge and skills. I also developed many soft skills. I learnt how to communicate with people. I was able to work in a team, under pressure and to cope with difficult situations.*

*(MSV4)*



*I have learnt more than I would during regular classes. On the intensive care unit, I was allowed to assist in some procedures that otherwise I could observe only after specialization. I also performed some procedures done typically by physicians, I would never do these as a nurse.*

*(MSV13)*



*It helped me to develop compassion for the sick and dying patients and improved my communication skills. Now when we are on the ward during the classes I find it much easier to deal with toilet issues, wash the patient or change their diaper.*

*(MSV14)*


Finally, volunteering was a chance for interprofessional collaboration, which helped MSVs to understand the peculiarities of other medical professions. Working with students from other faculties helped them “leave their own boxes”, “see them from the outside”, and appreciate the knowledge and skills of other healthcare professionals. This, in turn, strengthened the “professional integration” and “sense of cohesion”. Some also acknowledged that volunteering made them realize how the system works in critical situations.


*As a future physician I could observe and work with other healthcare professionals, i.e., nurses, and know their peculiarities. (…) It strengthened a solidarity between healthcare professionals.*

*(MSV4)*



*I could observe how the system works during the pandemic, how do hospitals work during this new situation. I could watch how the ward operates and how nurses, paramedics, physicians and technicians work.*

*(MSV8)*


Simultaneously, MSVs admitted that volunteering during the pandemic was a source of serious burden. However, although all respondents were afraid of being infected and bringing the virus to their homes, it was especially students who worked with patients, either in hospitals, emergency units, or sanitary-epidemiological stations, who complained about the high pressure and stress they were dealing with, and stressed the emotional strain and “moral injury” which made them physically and emotionally exhausted. Especially, that psychological burden, feelings of helplessness, guilt, and frustration were coupled with the absence of psychological support.


*The hardest thing was conversations. Sometimes it was difficult to forget about them. I kept them in my head and I often thought that I should have said something else or differently. (…) I remember one conversation with a mother of a teenage girl who struggled with self-destructive behaviours: scarification, the ‘S thoughts’, overdosing paracetamol [silence]. This was the worst call. I was shaken emotionally and I reminiscenced about it for a long time and had flashbacks [silence]. After such calls I needed a day or two off to go through it.*

*(MSV6)*



*The most difficult were conversations with the patients. Sometimes I felt helpless, as no one knew back then what to do, how to respond to questions… We were so confused.*

*(MSV10)*



*Sometimes I felt irritated because there were so many patients and due to staff shortages they were not cared for enough. It was very frustrating because I often couldn’t devote them more time and answer all their questions. I often left the zone embittered, irritated and felt sorry for the patients.*

*(MSV5)*


Although all MSVs claimed that the tasks performed during their service were adjusted to their qualifications, many complained over a lack of social skills and stressed how unprepared they felt for dealing with offensive or aggressive patients. Moreover, while complaining about the lack of a protocol on how to deal with such situations, students stressed how difficult and emotionally challenging it was to talk to patients.


*Especially during the first month patients were very against us. They associated us with restrictions in entering hospital and called us many bad words. We faced many negative emotions, especially when they gathered together, were tired and didn’t understand why we let some persons in and others not. Frequently we met with insults.*

*(MSV16)*



*Although during the first year we had some humanities, pedagogy or psychology, the studies didn’t prepare me for that: meeting people face-to-face, when they were often mad or aggressive while waiting for such a long time at the front of the hospital. Sometimes it was a bit dangerous.*

*(MSV15)*


Some MSVs felt unappreciated or suggested that students could have been used better. Especially medical students and those in their final years stressed that they could have performed more demanding tasks so they could relieve the staff.


*Some of us could have been used better. Being a PhD student myself, I graduated from biotechnology and I am familiar with all those laboratory things like isolation, and PCR. I would be more useful in the COVID Lab.*

*(MSV7)*



*I still have this feeling that we were not being used properly. There were hundreds of volunteers, but they didn’t recognize properly our potential and didn’t used us properly. People were very eager to work but our potential was wasted.*

*(MSV16)*


Among other burdens, MSVs stressed that constant performing of the same simple tasks was monotonous and boring. Others found it difficult to reconcile volunteering and studying. Some also admitted that volunteering had a negative impact on their social relations.


*After performing the same tasks for two months I was a bit tired and bored. It was so monotonous and non-developing.*

*(MSV1)*



*The hardest thing was to reconcile volunteering and studies.*

*(MSV19)*



*It negatively affected my social relations.*

*(MSV16)*


#### 3.2.7. Theme 7: Social Perception of MSV

MSVs reflected on how they were perceived by others and what reactions they faced during their voluntary service. Most respondents admitted receiving positive feedback from their families, including pride and support. However, at the beginning, parents and relatives were often ambivalent about it and tried to discourage them.


*My family was very supportive and proud. However, my extended family was like: ‘Why do you need it’, ‘Give it a break’. They were telling me I risk a lot and that I put relatives at risk. They discouraged me. But it was only at the beginning. Later they were proud and supportive.*

*(MSV2)*



*My entire family was against it. My parents were afraid and they tried to discourage me. Now they are proud and I have become a ‘family star’.*

*(MSV8)*


Some students mentioned positive reactions from the university authorities who acknowledged the importance of their service.


*On the University’s web page I saw videos describing our work. It stressed that not only professionals work in the laboratory but also the volunteers. I saw their gratitude. I felt appreciated and it made me very happy.*

*(MSV17)*



*A dean came to us several times to see how we work and to show his appreciation. We even received an email from the hospital’s director who acknowledged our work.*

*(MSV16)*


Moreover, the healthcare professionals also appreciated the students’ help, showing their support and respect. Others recalled the gratitude expressed by patients and their families.


*Healthcare professionals were very supportive and they often expressed their gratitude and told us how they appreciate our presence and help.*
*They often called us ‘heroes’.*

*(MSV1)*



*Medical personnel appreciated us a lot. They were very thankful and stressed that thanks to us those, who worked at the front line were much safer. Nurses and receptionists were calling us ‘our angels’.*

*(MSV16)*


Although similar positive reactions were expressed by students who did not volunteer, it was the peers, including university colleagues, who were often sceptical about the whole idea of students’ volunteering. Thus, although some respondents were admired by their friends, most MSV suggested that peers’ reactions were often negative and discouraging: some believed that volunteering was irresponsible as they might pose a risk to others, while others argued that volunteers were being used as a cheap workforce. Consequently, MSVs complained about the hate on the Internet and did not understand why some medical students were so aggressive about it.


*My friends from the university who didn’t know that I had volunteered were saying it was for the idiots; that one might get infected, that there was no equipment, one wouldn’t get paid and that it was not worth it as they wouldn’t let us do anything important.*

*(MSV8)*



*My colleagues tried to discourage me and said I shouldn’t do it; and on Facebook ‘the swamp’ spilled. I saw the discussion on the web saying that they were trying to use us as a cheap workforce. They were mad at the University that it asked us to volunteer. It really irritated me. I couldn’t understand their need to comment on my decision and being so negative and aggressive about it.*

*(MSV3)*


For some students, volunteering was a source of discriminatory behaviours, including social exclusion within the student community, from teachers, or neighbours. Even though they perceived such prejudices as “small things”, still they felt stigmatized, marginalized, and discriminated against. They stressed that being perceived as an ‘infection risk’ and a threat rather than as individuals who were offering their service to society made them feel sad and disappointed.


*Once, when I returned to my apartment after a twelve or eighteen hours of service I found a note stuck in the door saying that my neighbours were asking me to leave my apartment immediately because I was a possible ‘source of the plague’. I felt bitterness and disbelief.*

*(MSV8)*



*In the summer time, when I was wondering whether I should come to a party, some friends told me ‘No, no, you don’t have to come. We know you are at work, you know’. It was such small things. Also some teachers, at another university where I also study, who knew about it, were afraid of letting me into the classes.*

*(MSV10)*


## 4. Discussion

The COVID-19 pandemic has resulted in the increased demand for healthcare workers worldwide. A similar situation occurred in Poland, which for years has had difficulty in retaining its health professionals and struggles with an inadequate number of practicing physicians or nurses. Consequently, the question of whether students’ involvement in the pandemic should be allowed or restricted provoked heated debate [1,3,6,7,8,9,11,20,21,22,23,24,26,27,28,29,30]. However, this has left many students disappointed and frustrated because they felt prepared and were eager to contribute to the patient care [19,25]. They argued that students have clinical knowledge and valuable skills that can be appropriately used to help physicians, nurses, administrative staff, and patients. Moreover, it was also suggested that students did not have to work on the frontlines caring for patients but could perform non-COVID-19 related tasks, such as supporting healthcare professionals in providing patient care in emergency rooms and hospital wards, interviewing patients, caring for outpatients through telemedicine, helping in administrative matters, calling patients with lab results, sewing protective masks, or providing childcare for healthcare workers [6,13,14,23,42,43,44]. This was important because, while those tasks did not pose a high risk of COVID-19 transmission, they helped to relieve the healthcare system.

Interestingly, although previous studies in the United States [6,23,25], the United Kingdom [19,30], Ireland [40], Denmark [10,11], Germany [12], Nigeria [45], Saudi Arabia [46], China [47], Vietnam [17], and Indonesia [18] reported high rates of willingness to volunteer among medical students, such willingness was not always followed by students’ readiness to practice [18,48]. Meanwhile, this study confirms findings from previous Polish studies [13,14,15,16] that suggest that the majority of students across various faculties and years expressed a strong interest in active participation during the current healthcare emergency. Additionally, it shows that volunteers were mainly motivated by a strong sense of altruism [6,9,18,47] and had felt a moral and professional responsibility to help during the pandemic [12,37,48]. For example, Danish medical students were primarily motivated by the desire to help and wanted to take pride in contributing [29], and Saudi students were driven by patriotism and moral values [46]. Similarly, while for many Polish medical students’ personal enhancement and growth or the desire to develop personal career were important motivators, the vast majority volunteered out of a sense of civic responsibility and were mainly driven by altruism and the ethical imperative to serve their community, their fellow healthcare professionals, and their patients [13,14,15,16]. Indeed, most respondents emphasized the humanitarian nature of medicine and referred to such values as altruism, courage, and a sense of duty. Thus, although the primary role of the volunteering was to relieve the healthcare system, it also reinforced such important medical values as altruism, public service, and professional solidarity. Consequently, this study shows that students’ commitment during the COVID-19 pandemic was a great example of active citizenship and civic responsibility: volunteering helped to cultivate a spirit of benevolence among these future healthcare professionals and was a chance to engage in positive behaviour [47]. Significantly, although students consulted their decision to engage in the fight against the pandemic with their teachers, families, or friends, none of the respondents indicated the ‘family’ or ‘colleague effect’. On the contrary, they all declared it was their personal and autonomous choice.

Interestingly, students pointed to various benefits stemming from volunteering, which included the normative ones (being useful for the community, participating in something important, fulfilling the civic duty, public service, and helping others), personal enhancement (psychological development and personal growth, realization of one’s passion, or having a sense of pride), career (learning new knowledge and skills, building and developing personal career, gaining professional experience, making useful contacts, interprofessional collaboration, or obtaining the academic credit) and social (new relationships with others, making new friends, working with other people, and respect from others). However, most respondents suggested that the pandemic was a unique teaching moment and active involvement in volunteering created new learning opportunities, especially in the field of competency-based medical education and project problem-based learning [11,39,49,50,51,52]. Indeed, in addition to altruistic and value-driven motivations, most participants recognized that the pandemic gave them a chance to gain new knowledge and improve their social, organisational, and stress management skills, and helped them to understand how the health system works as a multidisciplinary whole [14]. Moreover, working together with students from various faculties was also a lesson for future interprofessional collaboration [43]. Thus, although Cullum et al. [53] suggested that the suspension of clinical placements during the pandemic may negatively influence the development of professional identity among medical students, this study supports the arguments of Findyartini et al. [54] that allowing students to play an active role during the outbreak contributed positively to the formation and internalization of their professional identity. Many respondents admitted that volunteering made them realize that although they were still students, they were already thinking, feeling, and acting as healthcare professionals. Consequently, they felt that by making a contribution to patient care, they have become part of a of healthcare team.

Simultaneously, in accord with Kalet et al. [24] this study suggests that while students’ engagement was voluntary, many experienced various forms of external pressure or a sense of coercion from their academic teachers, fellow students, or society. Even though such pressure was mainly present at the beginning of the pandemic, Shibu [55] showed that also during the second wave, when students had returned to university, they often felt obliged to continue their volunteering due to either internal motivation or the external pressures of the healthcare environment.

What is also important is that most respondents experienced a mix of an obligation to join in the fight against the COVID-19 pandemic, and fear [14,25,56,57]. Interestingly, students mainly expressed fear over the possibility of transmitting the infection to others in their homes and worried that the pandemic might negatively affect their studies or that the healthcare system might collapse [15,16,18,22,52]. Moreover, in accord with the observation made by Bazan et al. [13] while the highest level of students’ fear was observed during the first wave of the pandemic, when the recruitment process begun it decreased over the course of volunteering. At the same time, although gender, age, or the faculty did not affect MSV’s responses, female students were more willing to volunteer for altruistic reasons as compared to male counterparts. On the other hand, they were more prone to develop anxiety and stress and were more affected by the fear of spreading the infection to others.

Another important finding is that the major concern regarding volunteering was related to perceived insufficient medical knowledge and skills [48]. Especially younger students felt insecure and were concerned over not being prepared for dealing with the pandemic. Simultaneously, most students lacked social skills—so important in dealing with offensive or aggressive behavior. Consequently, MSVs complained about the lack of university courses on interpersonal communication, especially in crisis situations. What is also worrying is that although most MSVs expressed generally positive opinions on the organization of volunteering, many still complained over the lack of access to psychological support and suggested that they were not cared for enough at the emotional level [13,14]. However, Spanish students also complained about not having received specific training or psychological support [22]. Finally, it was also alarming that there were students who reported being the target of stigmatization, prejudice, and discriminatory behavior related to volunteering [13].

Although respondents emphasized that their involvement in voluntary service during the pandemic had received positive feedback, they mentioned that many students who did not volunteer felt they were being exploited as a free workforce. Moreover, although all participants emphasized the voluntary character of their service, some suggested that as it lasted over time and the jobs they performed were ordinarily waged, they should be drafted a formal employment contract and have regular payment. Similar concerns were found among Danish students who believed that for many receiving a salary would be an important additional motivational factor [29]. Similarly, while 56.7% of Nigerian nursing students suggested that the monetary reward would be a good motivation [45], more than half of British respondents believed that they should be offered a formal contract of employment and legal protection whilst working in a clinical capacity [30].

## 5. Strengths and Limitations

This study has some limitations. First, only twenty-one MSVs were interviewed. However, despite the small sample, thematic saturation was achieved. Second, the study has a local dimension as only responses from one Polish medical university were analysed. Consequently, it would be desirable to compare the findings from other locations in the country. Third, because five students resigned, the results represent solely the opinions of those MSVs who agreed to participate in the study and cannot be generalised for the entire population of MSVs either in Poznan or in Poland as a whole. Fourth, there was a predominance of female participants (F:M 16:5), which limits the results’ transferability to male students, although this imbalance is representative of the female predominance within the medical studies in Poland. Finally, as the analysis was performed by one author alone, there was a higher risk of subjectivity that might have influenced both the choice of the themes and the interpretation of the data.

However, despite these limitations, some advantages of this study should also be acknowledged. Most importantly, as there is a scarcity of previous work on the topic, this research fills the gap in the literature regarding the experiences of MSVs facing the COVID-19 pandemic in Poland. Indeed, while previous studies discussed students’ attitudes, motivations, or willingness to volunteer, to the best of my knowledge this is the first study that explores students’ lived experiences during the COVID-19 pandemic in Poland. Additionally, while previous studies often focused on medical students, this research highlights the experiences of students from various faculties enrolled in an interprofessional project. Moreover, it describes the experiences of not only final year students but also those in their early years of training, who are typically not immersed in clinical settings. Thus, while emphasizing the role of students during the current health crisis, this study provides new insights into how individualized the students’ response to COVID-19 was. Finally, it enabled the MSVs to share their experiences, which might have a therapeutic value.

## 6. Conclusions

This study suggests that although medical students’ involvement in volunteering during the COVID-19 pandemic was intended to prevent shortages of healthcare workers and relieve the system before it reached a personnel crisis, it was also a self-satisfying and career-related experience. Although being committed gave students the feeling of taking part in the fight against the pandemic, MSVs emphasized that in times when medical universities were closed and classes were held online, students’ volunteering became an important part of the service learning and created an opportunity for education. Thus, while it benefited students, patients, and the healthcare system, students’ involvement reinforced such important values of medical ethos as: altruism, public service, and (professional) solidarity.

However, although this research confirms the view that medical students are willing to contribute in the response to health disasters and emergencies and that their involvement helped to relieve the healthcare system, it also shows that in order to promote the idea of students volunteering, improve the healthcare’s organization, enhance the students’ safety, and better prepare them for next health crisis, some systemic approaches should be undertaken:To ensure that medical students can effectively volunteer in future disasters, policymakers and university authorities should not overlook the potential force of medical students as a support to the health system;There is an urgent need for promoting the idea of volunteering among all medical students;Students’ participation should always be voluntary, and free of any type of external, formal, or informal pressure;Students volunteering should be treated as a way of supporting the student learning process and incorporated into university curricula;Medical universities should integrate global health and disaster medicine to the medical curricula;There is a need for special preparation courses that would improve students’ communication skills and handling of difficult situations;Dedicated university courses on emergency decision making, coping, and leadership during a crisis should be integrated into the medical curricula;To minimize the risk of litigation, students should always receive proper induction and be trained about their responsibilities, procedures, and protocols;All tasks performed by MSVs should be safe, adjusted to their capabilities and level of training, and performed under supervision and institutional medico-legal protection;When engaging MSVs, special attention should be paid to the risks of infecting the patients and students as well as to the PPE shortages;Because MSVs may perform some tasks remotely, telemedicine should be further developed;Because volunteering during the pandemic was physically and emotionally challenging, MSVs must be provided with mental health support in physically or emotionally challenging roles.

## Figures and Tables

**Figure 1 ijerph-19-02314-f001:**
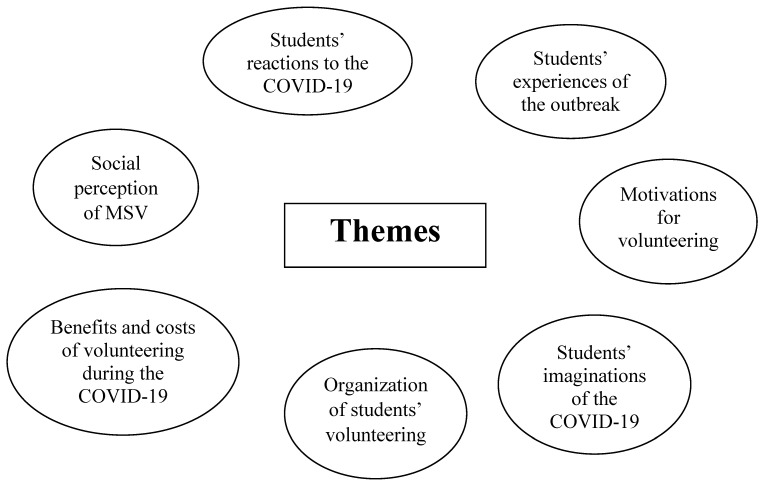
Themes Emerging from Qualitative Analysis.

**Table 1 ijerph-19-02314-t001:** Study participants.

No	Code	Gender	Faculty	Year of Study (While Volunteering)	Time Spent Volunteering	Tasks Performed	Earlier Volunteering
1.	MSV1	female	medicine	5	2 months	triage in the hospital	yes
2.	MSV2	female	medicine	1	3 months	triage, nurse assistance	yes
3.	MSV3	male	medicine	2	5 months	tele-information in the hospital	yes
4.	MSV4	female	public health	4	1.5 months	telephone advice in a sanitary-epidemiological station	yes
5.	MSV5	male	rescue medicine	2	12 months	emergency room	yes
6.	MSV6	female	public health	1	2 months	tele-information	yes
7.	MSV7	female	medicine	2 (PhD student)	1.5 months	drive-thru testing	yes
8.	MSV8	male	rescue medicine	1	3 months	pre-triage in the hospital and bringing packages from families to patients	no
9.	MSV9	female	pharmacy	4	2 months	drive-thru testing	yes
10.	MSV10	female	nursing	2	3 months	call-centre in the hospital, triage in the hospital	yes
11.	MSV11	female	medicine	4	4 months	organization of PPE supplies; translation of English texts about COVID-19	yes
12.	MSV12	female	medical analytics	3	2,5 months	telephone advice in a sanitary-epidemiological station	yes
13.	MSV13	male	nursing	1&2	4 months	hospice, anestiosiology, intensive care ward	yes
14.	MSV14	female	nursing	1	5 months	hospice, non-COVID-19 wards	no
15.	MSV15	female	nursing	2	6 months	translation of English texts about COVID-19	yes
16.	MSV16	female	medicine	5	3 months	triage in the hospital	yes
17.	MSV17	female	medicine	3	1 month	the University Coronavirus Laboratory at PUMS (administrative work)	no
18.	MSV18	female	nursing	4	3 months	triage, nurse assistance	no
19.	MSV19	female	medicine	1	2 months	triage in the hospital	no
20.	MSV20	female	nursing	3	2.5 months	triage, nurse assistance	yes
21.	MSV21	male	medicine	3	3 months	tele-information in the hospital	yes

## Data Availability

The data and codes that produce the findings reported in this article are available from the corresponding author on reasonable request.

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
