# Peer review of "‘Who Else If Not We’. Medical Students’ Perception and Experiences with Volunteering during the COVID-19 Crisis in Poznan, Poland"

_ijerph, 2022, doi:10.3390/ijerph19042314_

Round 1

Reviewer 1 Report

This article clarified the perceptions and lived experiences of medical student volunteers and has important implications for understanding the role of medical students in the fight against the outbreak in Polish. At the same time, this article also has the following points that can be revised and improved.

1)  This paper needs to make more sense of what is important about focusing on the perceptions and lived experiences of medical student volunteers in Poland. Generally thinking, the perceptions and lived experiences of frontline physicians and health care workers against the outbreak will be more meaningful than those of student volunteers.

2)  The discussion and conclusion section does not allow the readers to understand the important findings and also does not reflect the originality of this study well. Therefore, it is recommended that the authors should state the originality of this study firstly in the introduction section while responding to them in the discussion section.

3)  The conclusion section should be more reflective of its policy implications. How the main results can be applied to the solution of a series of problems brought about by the increased pressure on the health care system due to the outbreak of the COVID-19.

4)  It is suggested that the authors need to add a description of the current characteristics of the COVID-19 crisis in Poznan and the overall situation of student volunteers, as well as clarify the specific positioning of Poznan in Poland in the material and methods section, thus making the results more general and generalizable.

5)  The title of this article can be revised as follows: ‘Who else if not we’. Medical Students’ Perception and Experiences with Volunteering During the COVID-19 Crisis in Poznan, Poland

Author Response

Dear Reviewer,

first of all, I would like to express my gratitude for giving me the opportunity to revise and resubmit my paper. I am indebted to your valuable suggestions and helpful comments. I hope that this revised paper is more consistent owing to your willingness to help.

As you will see from this cover letter, I have put a lot of effort into revising my paper in accord with your suggestions. I have been convinced by all your arguments and I am very grateful for pointing these things out. I believe that I have also answered all the questions you have raised. Still, should it happen that I have missed and/or misunderstood any vital comment, I would be more than happy to promptly rectify this and further revise my article.

Below, I detail the changes that I have made in accord with your suggestions and comments (please note that in the revised manuscript I have taken the liberty of marking the most important changes in colour to facilitate their checking).

1)This paper needs to make more sense of what is important about focusing on the perceptions and lived experiences of medical student volunteers in Poland. Generally thinking, the perceptions and lived experiences of frontline physicians and health care workers against the outbreak will be more meaningful than those of student volunteers.

I am grateful to for this remark, because it has helped me to clarify that while knowing the perceptions of the frontline healthcare workers is important, understanding medical students’ experiences with the pandemic is also of crucial importance as it may help us to guide future practice and prepare them better for the next health crisis. Especially that some students felt unprepared for dealing with the pandemic and for others volunteering during the COVID-19 was a source of serious burden.

This has been done on page 5 (lines 148-154). Thus, after revision it now says (the revised part has been marked in colour):

“Thus, this study explores medical students volunteers (MSV) perspective and their lived experiences of volunteering during the COVID-19 pandemic. Although knowing the perceptions of frontline healthcare workers is important, understanding medical students’ experiences with the pandemic is also of crucial importance as it may help us to guide future practice and prepare them better for next health crisis. Especially that some students felt unprepared for dealing with the pandemic and for others volunteering during the COVID-19 was a source of serious burden.”

I hope my explanation is sufficient and I am, again, very grateful for pointing this out.

2)The discussion and conclusion section does not allow the readers to understand the important findings and also does not reflect the originality of this study well. Therefore, it is recommended that the authors should state the originality of this study firstly in the introduction section while responding to them in the discussion section.

Having been persuaded by the remark, I have clarified that while earlier studies focussed on medical students’ attitudes, motivations and willingness to volunteer during the COVID-19 outbreak or the role of volunteering in the education of future healthcare professionals and formation of their professional identity, there is still a great shortage of research on the students’ lived experiences during the pandemic. Consequently, this study identifies the facilitators, barriers and factors affecting medical students’ satisfaction with volunteering during the coronavirus outbreak. Moreover, while most previous studies were designed as quantitative research, this in-depth study used qualitative methods which helped to better understand the students’ motivations, opinions and lived experiences. This has been done at the beginning of the ‘Methods’ section, on pages 5-6 (160-170). Thus, after revision it now says (the revised part has been marked in colour):

“While earlier studies focussed on medical students’ attitudes, motivations and willingness to volunteer during the COVID-19 outbreak or the role of volunteering in the education of future healthcare professionals and formation of their professional identity, there is still a great shortage of research on the students’ lived experiences during the pandemic. Consequently, this study identifies the facilitators, barriers and factors affecting medical students’ satisfaction with volunteering during the coronavirus outbreak. Moreover, while most previous studies were designed as quantitative research this study used qualitative methods [37]. Thus, such in-depth study may help us to better understand the students’ motivations, opinions and lived experiences. This in turn, may help us to give them the right foundation to learn, thrive and ensure that medical students will be better prepared when faced with another pandemic.”

I hope my explanation satisfies the concerns raised.

3) The conclusion section should be more reflective of its policy implications. How the main results can be applied to the solution of a series of problems brought about by the increased pressure on the health care system due to the outbreak of the COVID-19.

I am grateful for this remark as well because it has helped me to reflect more on the policy implications of medical students volunteering. This has been done on page 25 (lines 907-937). Thus, after revision it now says (the revised part has been marked in colour):

“However, although this research confirms the view that medical students are willing to contribute in the response to health disasters and emergencies and that their involvement helped to relieve the healthcare system, it also shows that in order to promote the idea of students volunteering, improve the healthcare’s organization, enhance the students’ safety and better prepare them for next health crisis, some systemic approach should be undertaken:

  1. to ensure that medical students can effectively volunteer in future disasters policymakers and university authorities should not overlook the potential force of medical students as a support to the health system;
  2. there is a urgent need of promoting the idea of volunteering among all medical students;
  3. students participation should always be voluntary, and free of any type of external, formal or informal, pressure;
  4. students volunteering should be treated as a way of supporting student learning process and incorporated into a university curricula;
  5. medical universities should integrate global health and disaster medicine to the medical curricula;
  6. there is a need for special preparation courses that would improve students’ communication skills and handling difficult situations;
  7. dedicated university courses on emergency decision making, coping and leadership during a crisis should be integrated into the medical curricula;
  8. to minimize the risk of litigation, students should always receive proper induction and be trained about their responsibilities, procedures and protocols;
  9. all tasks performed by SV should be safe, adjusted to their capabilities and level of training and performed under supervision and institutional medico-legal protection;
  10. when engaging MSV, special attention should be paid to the risks of infecting the patients and students as well as to the PPE shortages;
  11. because SV may perform some tasks remotely, telemedicine should be further developed;
  12. because volunteering during the pandemic was physically or emotionally challenging, MSV must be provided with mental health support in physically or emotionally challenging roles.”

I hope my explanation satisfies the concerns raised.

4) It is suggested that the authors need to add a description of the current characteristics of the COVID-19 crisis in Poznan and the overall situation of student volunteers, as well as clarify the specific positioning of Poznan in Poland in the material and methods section, thus making the results more general and generalizable.

In accord with the Reviewer’s apt suggestion I have added a paragraph describing the current characteristics of the COVID-19 crisis in Poznan and the overall situation of the student volunteers. This has been done on page 4-5 (lines 115-147). Thus, after revision it now says (the revised part has been marked in colour):

“Similar discussions occurred in Poland, where the first case of the COVID-19 infection was confirmed on March 4 2020. Similarly, the first case of the disease in Poznan was confirmed five days later, and it was the first COVID victim in the country who died on March 12. Consequently, starting from March 10 various lockdown-type control measures were imposed by the Polish government, including the closure of all universities which was announced on March 11. However, following a call from the Ministry of Health, Poznan University of Medical Sciences (PUMS) in collaboration with the university’s student organizations started inviting its students to volunteer. The reason for this being that although even before the pandemic Poland suffered from the healthcare workforce shortages, as in 2017 there were 2.4 doctors and 5.1. nurses per 1,000 population (OECD mean 3.5 and 8.8, respectively) [33], the COVID-19 has deepen the healthcare staffing crisis and the country’s healthcare system reached its limits. Indeed, from the beginning of the pandemic 152,436 healthcare professionals in Poland were infected (81,844 nurses, 32,872 physicians, 13,410 physiotherapists, 8,416 midwives, 4,616 pharmacists, 4,116 paramedics, 3,986 dentists, 3,146 laboratory diagnosticians and 30 feldshers) and 582 have died (251 physicians, 201 nurses, 61 dentists, 24 midwives, 22 pharmacists, 7 paramedics, 7 physiotherapists, 5 laboratory diagnosticians and 4 feldshers) [34]. Moreover, as of January 27 2022, in Poznan and the Wielkopolska region alone, the total number of infections reached 442,172 (12,66% of population of the region) and 9,610 people have died (9,19% of all infections and 8,93% of deaths in the country) [35]. Consequently, from the very beginning the outbreak of the coronavirus disease presented itself not only a public health emergency but also as a challenge to the health system.

Thus, on 13 of March 2020 the rector of PUMS and the governor of the Wielkopolska province signed an agreement which made it possible for volunteers to work in various health care facilities, including hospitals, emergency units, hospital pharmacies, sanitary-epidemiological stations and the university’s diagnostic laboratory. As a result of this, more than seven hundred students from the faculties of Medicine, Pharmacy and Health Sciences volunteered during the first wave of pandemic alone.

However, the problem was that in contrast to many other jurisdictions in the Polish law student volunteering is not regarded as a mandatory form of experiential learning such as internships. Consequently, although, medical students engage in various forms of volunteering medical universities in Poland do not regard student volunteering as a part of university curricula, nor it is treated as a way of supporting student learning process [36].”

I hope my explanation satisfies the Reviewer’s concerns. At the same time, I would like to point out that I tried not to exceed the world limit too much, as my manuscript is already rather long. Still, should it happen that I have missed and/or misunderstood any vital comment, I would be more than happy to promptly rectify this and further revise my article.

5) The title of this article can be revised as follows ‘Who else if not we’. Medical Students’ Perception and Experiences with Volunteering During the COVID-19 Crisis in Poznan, Poland

Having been persuaded by the objection, I have revised the title of my article accordingly. Thus, after revision it now says: ‘Who else if not we’. Medical Students’ Perception and Experiences with Volunteering During the COVID-19 Crisis in Poznan, Poland’.

Moreover, in accord with that change, the abbreviation used to describe study participants has been also revised. Thus, I now use MSV instead SV, through the whole text.

Reviewer 2 Report

This is very interesting article. I only have a few small comments:

1. "Because there is a paucity of research on students’ experiences during the pandemic, the study has been designed as a qualitative research [33]."  -  It is worth adding one more sentence that will explain this. It may not necessarily be clear to the reader why "paucity of research" results in the selection of qualitative research. 

2. "When the pandemic gained momentum I was sacred over the health of my loved ones, my family." -  Should be "scared".

3. "During the first weeks of the pandemic most SV were mainly focused on personal issues and worried about their education and professional futures."  - I propose to replace "the first weeks" with "during the first wave of the pandemic" because immediately after that is the quote "after two to three months". 

4. "Moreover, regardless of their age, gender, faculty, initial level of fear, the time of recruitment or time spent on volunteering, all SV expressed a high level of satisfaction from volunteering and stressed the countless benefits that arose from it." -  I'ts worth to write about it few words in the discussion part.

5. "Although during the first year we had same humanities, pedagogy or psychology..." - should be "some humanities" i think...

Author Response

Dear Reviewer,

first of all, I would like to express my gratitude for giving me the opportunity to revise and resubmit my paper. I am indebted to your valuable suggestions and helpful comments. I hope that this revised paper is more consistent owing to your willingness to help.

As you will see from this cover letter, I have put a lot of effort into revising my paper in accord with your suggestions. I have been convinced by all your arguments and I am very grateful for pointing these things out. I believe that I have also answered all the questions you have raised. Still, should it happen that I have missed and/or misunderstood any vital comment, I would be more than happy to promptly rectify this and further revise my article.

            Below, I detail the changes that I have made in accord with your suggestions and comments (please note that in the revised manuscript I have taken the liberty of marking the most important changes in colour to facilitate their checking).

  1. "Because there is a paucity of research on students’ experiences during the pandemic, the study has been designed as a qualitative research [33]." - It is worth adding one more sentence that will explain this.It may not necessarily be clear to the reader why "paucity of research" results in the selection of qualitative research.

I am very grateful for this remark because it has helped me to clarify that the reason for choosing

qualitative research. Thus, I explain that in contrast to quantitative research this in-depth study uses qualitative methods which may help to better understand the students’ motivations, opinions and lived experiences. This has been done on page 5-6 (lines 160-170). Thus, after revision it now says (the revised part has been marked in colour):

“While earlier studies focussed on medical students’ attitudes, motivations and willingness to volunteer during the COVID-19 outbreak or the role of volunteering in the education of future healthcare professionals and formation of their professional identity, there is still a great shortage of research on the students’ lived experiences during the pandemic. Consequently, this study identifies the facilitators, barriers and factors affecting medical students’ satisfaction with volunteering during the coronavirus outbreak. Moreover, while most previous studies were designed as quantitative research this study used qualitative methods [37]. Thus, such in-depth study may help us to better understand the students’ motivations, opinions and lived experiences. This in turn, may help us to give them the right foundation to learn, thrive and ensure that medical students will be better prepared when faced with another pandemic.”

I hope my explanation satisfies the concerns raised.

  1. "When the pandemic gained momentum I was sacred over the health of my loved ones, my family." - Should be "scared".

Apart from revising the indicated linguistic mistakes, the manuscript has been read and copy-edited by a British native speaker of English.

  1. "During the first weeks of the pandemic most SV were mainly focused on personal issues and worried about their education and professional futures." - I propose to replace "the first weeks" with "during the first wave of the pandemic" because immediately after that is the quote "after two to three months".

Having been persuaded by the objection, I have reformulated the aforementioned sentence. Thus, after revision it now says:

“During the first wave of the pandemic most SV were mainly focused on personal issues and worried about their education and professional futures.”

I hope my explanation is sufficient and I am grateful for this point.

  1. "Moreover, regardless of their age, gender, faculty, initial level of fear, the time of recruitment or time spent on volunteering, all SV expressed a high level ofsatisfaction from volunteering and stressed the countless benefits that arose from it." - I'ts worth to write about it few words in the discussion part.

I am very grateful for this remark because it has helped me develop the description of multiple benefits reported by the students, including professional development, learning, practicing communication skills, interprofessional education and professional identity formation. This has been done on pages 22-23 (lines 832-839). Thus, after revision it now says (the revised part has been marked in colour):

“Interestingly, students pointed to various benefits stemming from volunteering, which included the normative ones (being useful for the community, participating in something important, fulfilling the civic duty, public service and helping others), personal enhancement (psychological development and personal growth, realization of one’s passion or having a sense of pride), career (learning new knowledge and skills, building and developing personal career, gaining professional experience, making useful contacts, interprofessional collaboration or obtaining the academic credit) and social (new relationships with others, making new friends, working with other people, respect from others). However, most respondents suggested that the pandemic was a unique teaching moment and active involvement in volunteering created new learning opportunities, especially in the field of competency-based medical education and project problem-based learning [11,39,49-52].”

I hope my explanation satisfies the Reviewer’s concerns.

  1. "Although during the first year we had same humanities, pedagogy or psychology..." - should be "some humanities" i think...

As noted above, apart from revising all typographical errors the entire text has revised by a native speaker of English.

Reviewer 3 Report

  1. Would you like to analysis the student's perception based on different faculty or sex... 
  2. Can you to describe the job difference of the SV, different kind of job will have different perception toward the pandemic.
  3. Do you explore the motivation why these student's want to be the SV in the pandemic era? family or colleague effect?

Author Response

Dear Reviewer,

first of all, I would like to express my gratitude for giving me the opportunity to revise and resubmit my paper. I am indebted to your valuable suggestions and helpful comments. I hope that this revised paper is more consistent owing to your willingness to help.

As you will see from this cover letter, I have put a lot of effort into revising my paper in accord with your suggestions. I have been convinced by all your arguments and I am very grateful for pointing these things out. I believe that I have also answered all the questions you have raised. Still, should it happen that I have missed and/or misunderstood any vital comment, I would be more than happy to promptly rectify this and further revise my article.

            Below, I detail the changes that I have made in accord with your suggestions and comments (please note that in the revised manuscript I have taken the liberty of marking the most important changes in colour to facilitate their checking).

  1. Would you like to analysis the student's perception based on different faculty or sex…

I am very grateful for this remark because it has helped me to clarify that although the study sample was small some differences related to participants sex and age were observed. Thus, I stress that while females seemed to be more willing to volunteer for altruistic reasons, they were also more prone to develop anxiety and stress. Finally, females were more affected by the fear of spreading the infection to others. On the other hand, especially younger students felt insecure and were concerned about not being prepared for dealing with the pandemic. This has been done on page 24 (lines 870-879). Thus, after revision it now says (the revised part has been marked in colour):

“What is also important is that most respondents experienced a mix of an obligation to join in the fight against the COVID-19 pandemic, and fear [14,25,56,57]. Interestingly, students mainly expressed fear over the possibility of transmitting the infection to others in their homes and worried that the pandemic might negatively affect their studies or that the healthcare system might collapse [15,16,18,22,52]. Moreover, in accord with the observation made by Bazan at al. [13] while the highest level of students’ fear was observed during the first wave of the pandemic, when the recruitment process begun it decreased over the course of volunteering. At the same time, although gender, age or the faculty did not affect MSV’s responses, female students were more willing to volunteer for altruistic reasons as compared to male counterparts. On the other hand, they were more prone to develop anxiety and stress and were more affected by the fear of spreading the infection to others.

Another important finding is that the major concern regarding volunteering was related to perceived insufficient medical knowledge and skills [45]. Especially younger students felt insecure and were concerned about not being prepared for dealing with the pandemic. Simultaneously, while some participants felt unprepared for dealing with the pandemic, most lacked social skills – so important in dealing with offensive or aggressive behaviours.”

I hope my explanation satisfies the Reviewer’s concerns. At the same time, again I am very grateful for pointing this out.

  1. Can you to describe the job difference of the SV, different kind of job will have different perception toward the pandemic.

Having been persuaded by this suggestion I have added a sentence explaining that especially students who worked with patients, either in hospitals, emergency units or sanitary-epidemiological stations complained over the high pressure, the psychological burden, feelings of helplessness, guilt and frustration. This has been done on page 18 (lines 649-655). Thus, after revision it now says:

“Simultaneously, MSV admitted that volunteering during the pandemic was a source of serious burden. However, although all respondents were afraid of being infected and bringing the virus to their homes, it was especially students who worked with patients, either in hospitals, emergency units or sanitary-epidemiological stations, who complained about the high pressure and stress they were dealing with, and stressed the emotional strain and “moral injury” which made them physically and emotionally exhausted. Especially, that psychological burden, feelings of helplessness, guilt and frustration were coupled with the absence of psychological support.”

Additionally, I have referred to this matter in the ‘Discussion’ section where I conclude that although all respondents expressed a high level of satisfaction from volunteering and stressed the countless benefits that arose from it, especially younger students and those who had contact with the patients experienced emotional burden, caused both by the fear of being infected or the possibility of passing the disease to others. Moreover, MSV complained about lack of knowledge required to manage some tasks or lack of social skills that would help them in dealing with offensive or aggressive behaviours. This has been done on page 24 (lines 875-879). Thus, after revision it now says:

“Especially younger students felt insecure and were concerned over not being prepared for dealing with the pandemic. Simultaneously, most students lacked social skills – so important in dealing with offensive or aggressive behaviours. Consequently, MSV complained over the lack of university courses on interpersonal communication, especially in crisis situations.”

I hope my explanation is sufficient. At the same time, I am indebted for pointing this out.

  1. Do you explore the motivation why these student's want to be the SV in the pandemic era? family or colleague effect?

While the entire theme 3 is dedicated to the medical students’ motivations for volunteering in accord with the Reviewer’s suggestion I have extended the analysis in this matter. Thus, I stress that although for many respondents personal enhancement and growth or the desire to develop personal career were important motivators, as they wanted to learn new skills and gain professional experience, the vast majority volunteered out of a sense of civic responsibility and their involvement was mainly driven by altruism and the ethical imperative to serve their community, their fellow healthcare professionals and their patients. Thus, I emphasize that although the prime role of the volunteering was to relieve the healthcare system, it also reinforced such important medical values as altruism, public service and professional solidarity.

            Simultaneously, I explain that although students consulted their decision to engage into voluntary service with their teachers, families or friends none of the respondents indicated to the ‘family’ or ‘colleague effect’. On the contrary, they all declared it was their personal and autonomous choice.

This has been done in the ‘Discussion section’ (page 22, lines 817-831). Thus, after revision it now says (the revised part has been marked in colour):

“Similarly, while for many Polish medical students personal enhancement and growth or the desire to develop personal career were important motivators, the vast majority volunteered out of a sense of civic responsibility and were mainly driven by altruism and the ethical imperative to serve their community, their fellow healthcare professionals and their patients [13–16]. Indeed, most respondents emphasized the humanitarian nature of medicine and referred to such values as altruism, courage and sense of duty. Thus, although the prime role of the volunteering was to relieve the healthcare system, it also reinforced such important medical values as altruism, public service and professional solidarity. Consequently, this study shows that students commitment during the COVID-19 pandemic was a great example of active citizenship and civic responsibility: volunteering helped to cultivate a spirit of benevolence among these future healthcare professionals and was a chance to engage in positive behaviour [48]. Significantly, although students consulted their decision to engage in the fight against the pandemic with their teachers, families or friends none of the respondents indicated the ‘family’ or ‘colleague effect’. On the contrary, they all declared it was their personal and autonomous choice.”

I hope my explanation satisfies the Reviewer’s concerns. At the same time, again I am very grateful for this valuable question.

Reviewer 4 Report

The article describes results of a study conducted to explores in a small sample of student volunteers perspective and experiences during the COVID-19 pandemic in Poland. The aim of the manuscript is interesting although it reports results of a study conducted in a very small sample of twenty-one students. To improve the manuscript, I suggest to discuss and give a more critical judgement on possible application of the results of the study.

Furthermore, I suggest the following points:

  • English revision is recommended.
  • The section Methods should report further details on the development and validation of the questionnaire
  • The section 3.2 Findings is too long and I suggest to summarize the main results.

Author Response

Dear Reviewer,

first of all, I would like to express my gratitude for giving me the opportunity to revise and resubmit my paper. I am indebted to your valuable suggestions and helpful comments. I hope that this revised paper is more consistent owing to your willingness to help.

As you will see from this cover letter, I have put a lot of effort into revising my paper in accord with your suggestions. I have been convinced by all your arguments and I am very grateful for pointing these things out. I believe that I have also answered all the questions you have raised. Still, should it happen that I have missed and/or misunderstood any vital comment, I would be more than happy to promptly rectify this and further revise my article.

            Below, I detail the changes that I have made in accord with your suggestions and comments (please note that in the revised manuscript I have taken the liberty of marking the most important changes in colour to facilitate their checking).

The article describes results of a study conducted to explores in a small sample of student volunteers perspective and experiences during the COVID-19 pandemic in Poland. The aim of the manuscript is interesting although it reports results of a study conducted in a very small sample of twenty-one students.

I am grateful for this remark. However, I would like to explain that while only twenty one MSV were interviewed, thematic saturation was achieved. Although there is an ongoing discussion on “how many” interviews “are enough,” the vast majority of authors suggest anywhere from 5 to 50 participants as adequate. Moreover, it is widely suggested that the thematic saturation should be achieved after conducting 13-16 interviews.

I hope my explanation is sufficient. At the same time, I am grateful for pointing this out.

To improve the manuscript, I suggest to discuss and give a more critical judgement on possible application of the results of the study.

I am grateful for this remark as well because it has helped me to reflect more on the policy implications of medical students volunteering. This has been done on page 25 (lines 907-937). Thus, after revision it now says (the revised part has been marked in colour):

“However, although this research confirms the view that medical students are willing to contribute in the response to health disasters and emergencies and that their involvement helped to relieve the healthcare system, it also shows that in order to promote the idea of students volunteering, improve the healthcare’s organization, enhance the students’ safety and better prepare them for next health crisis, some systemic approach should be undertaken:

  1. to ensure that medical students can effectively volunteer in future disasters policymakers and university authorities should not overlook the potential force of medical students as a support to the health system;
  2. there is a urgent need of promoting the idea of volunteering among all medical students;
  3. students participation should always be voluntary, and free of any type of external, formal or informal, pressure;
  4. students volunteering should be treated as a way of supporting student learning process and incorporated into a university curricula;
  5. medical universities should integrate global health and disaster medicine to the medical curricula;
  6. there is a need for special preparation courses that would improve students’ communication skills and handling difficult situations;
  7. dedicated university courses on emergency decision making, coping and leadership during a crisis should be integrated into the medical curricula;
  8. to minimize the risk of litigation, students should always receive proper induction and be trained about their responsibilities, procedures and protocols;
  9. all tasks performed by SV should be safe, adjusted to their capabilities and level of training and performed under supervision and institutional medico-legal protection;
  10. when engaging MSV, special attention should be paid to the risks of infecting the patients and students as well as to the PPE shortages;
  11. because SV may perform some tasks remotely, telemedicine should be further developed;
  12. because volunteering during the pandemic was physically or emotionally challenging, MSV must be provided with mental health support in physically or emotionally challenging roles.”

I hope my explanation satisfies the concerns raised.

Furthermore, I suggest the following points:

  1. English revision is recommended.

In accord with this suggestion the manuscript has been read and copy-edited by a native speaker of English.

  1. The section Methods should report further details on the development and validation of the questionnaire.

I am very grateful for this remark because it has helped me to clarify the development and validation of the questionnaire. Thus, in accord with this suggestion I have added paragraph describing the Qualitative Pretest Interview (QPI) approach that was used. This has been done on page 6 (lines 175-198). Thus, after revision it now says (the revised part has been marked in colour):

“Thus, semi-structured interviews with students who volunteered during the COVID-19 pandemic in Poznan, Poland, were performed. Because the study focussed on the students’ lived experiences, the meanings they gave to those experiences and the choices they made based on those meanings, an interpretative phenomenological approach was used for this study [34]. The questionnaire was developed according to the Qualitative Pretest Interview (QPI) approach [39]. Thus, the initial list of the interview questions was developed after a thorough analysis of academic literature on the medial students’ volunteering during the COVID-19 pandemic. The structure of the questionnaire was constructed in consideration of the study aim and focussed on medical students’ lived experiences of volunteering during the outbreak. Thus, the interview questionnaire, which consists of 7 categories of questions, was designed to find out what meanings students gave to their experiences as volunteers and how theses meanings influenced their decisions and choices: what were students’ reactions after hearing about the COVID-19 pandemic and the governmental restrictions, why they decided to engage in volunteering and what were the motivations behind their decision, what did they do during their voluntary service, what were their experiences with volunteering during the pandemic, how they rated organization of students’ volunteering, and what reactions they faced during their voluntary service. Thus, the questionnaire consisted of 32 questions, which facilitated the identification of specific issues related to medical students’ volunteering during the current healthcare crisis (Appendix 1).

Before carrying out the formal phase of qualitative research, a series of three pretest interviews were conducted to assess the instrumentation rigor and to formulate measures to address any limitations or threats to bias and management procedures. While it helped to reformulate four questions, it also enabled me to identify various obstacles and increase the methodological and social reliability of the questionnaire that are central to any qualitative research. The final version of the questionnaire was evaluated by an three external reviewers: two medical students and one sociologist and received approval from the University Student Council Board (USCB). Additionally, ethics approval and research governance approval were obtained from the PUMS Bioethics Committee (KB – 831/20).”

I hope my explanation addresses the Reviewer’s concerns. I am very grateful for pointing this out.

  1. The section 3.2 Findings is too long and I suggest to summarize the main results.

Having been persuaded by the objection, I have tried to compress the findings and summarize the main results. At the same time I had to elaborate on some themes that the other Reviewer pointed to. I therefore tried to strike a happy balance between these two demands.

Round 2

Reviewer 1 Report

  1. While the authors point out the fact that there is less research focusing on students' lived experiences, this still does not reflect well on the originality of this paper. The paucity of research may also be a result of the low value of the research. Therefore, the authors need to point out how focusing on students' lived experiences can solve the urgent problems faced by the Polish healthcare system during the epidemic? This point needs to be added further.
  2. The newly added paragraph (lines 102-106) feels less relevant to the current characteristics of the COVID-19 crisis in Poznan and can be considered for deletion.
  3. Please provide 2-3 key policy implications based on the key findings of this study in the section on lines 719-746. It currently feels like a list of content that does not reflect a strong relationship to the important findings of this study.
  4. Please correct SV in Figure 1 to MSV.

Reviewer 3 Report

More comprehensive description on result process

Using qualitative interview to point the unmet need during pandemic but only focus on small amount of students.  How to proof the unmet need is worth to improve before the second pandemic or if the pandemic persisted?